# Designing the Next Generation of Fe⁰-Based Filters for Decentralized Safe Drinking Water Treatment: A Conceptual Framework

**Huichen Yang [1], Rui Hu [2], Arnaud Igor Ndé-Tchoupé [3], Willis Gwenzi [4], Hans Ruppert [5],\*
and Chicgoua Noubactep [2,\*]**

[1]   Angewandte Geologie, Universität Göttingen, Goldschmidtstraße 3, D-37077 Göttingen, Germany;
      huichen.yang@geo.uni-goettingen.de
[2]   School of Earth Science and Engineering, Hohai University, Fo Cheng Xi Road 8, Nanjing 211100, China;
      rhu@hhu.edu.cn
[3]   Department of Chemistry, Faculty of Sciences, University of Douala, Douala B.P. 24157, Cameroon;
      ndetchoupe@gmail.com
[4]   Biosystems and Environmental Engineering Research Group, Department of Soil Science and Agricultural
      Engineering, University of Zimbabwe, Mt. Pleasant, Harare MP167, Zimbabwe; wgwenzi@agric.uz.ac.zw
[5]   Department of Sedimentology & Environmental Geology, University of Göttingen, Goldschmidtstraße 3,
      D-37077 Göttingen, Germany
\*   Correspondence: hrupper@gwdg.de (H.R.); cnoubac@gwdg.de (C.N.)

**Abstract:** The ambitious United Nations Sustainable Development Goal for 2030 to "leave no one behind" concerning safe drinking water calls for the development of universally applicable and affordable decentralized treatment systems to provide safe drinking water. Published results suggest that well-designed biological sand filters (BSFs) amended with metallic iron (Fe⁰-BSFs) have the potential to achieve this goal. Fe⁰-BSFs quantitatively remove pathogens and a myriad of chemical pollutants. The available data were achieved under various operating conditions. A comparison of independent research results is almost impossible, especially because the used Fe⁰ materials are not characterized for their intrinsic reactivity. This communication summarizes the state-of-the-art knowledge on designing Fe⁰-BSFs for households and small communities. The results show that significant research progress has been made on Fe⁰-BSFs. However, well-designed laboratory and field experiments are required to improve the available knowledge in order to develop the next generation of adaptable and scalable designs of Fe⁰-BSFs in only two years. Tools to alleviate the permeability loss, the preferential flow, and the use of exhausted filters are presented.

**Keywords:** filter beds; pathogen removal; point-of-use technologies; safe drinking water; zero-valent iron

---

## 1. Introduction

The continuous pollution of natural water is a serious global problem. Water pollution affects human and animal health, ecosystems, fishing, recreation, transportation, and other commercial activities [1,2]. Severe water pollution is mainly of anthropogenic origin, but geogenic pollution (e.g., As, F, U) is also prevalent in many regions of the world [3,4]. In addition, water pollution by pathogens is a universal problem [5,6]. There are several well-known water-related human diseases associated with unsafe drinking water, including cholera, arsenosis, and dental/skeletal fluorisis. Therefore, the international community has been searching for affordable, applicable, and efficient solutions for safe drinking water supply for many decades [2,4,6–9].

In September 2015, the United Nations (UN) adopted the Sustainable Development Goals (SDGs) to improve the human condition substantially by 2030. Two of the 17 UN SDGs focus explicitly on water: (i) Goal 6 "Ensure availability and sustainable management of water and sanitation for all", and (ii) Goal 14 "Conserve and sustainably use the oceans, seas and marine resources for sustainable development" [4,10]. The present communication is focused on Goal 6, which basically calls for improving water quality to supply the world's population with safe drinking water by 2030. Hering et al. [10] posited that fulfilling this UN SDG within 15 years requires that "we effectively translate existing knowledge into practical solutions". Actually, only 10 years are left and it is a good idea to review the achievements and shape the research activities for the coming years. In the current study, this is achieved by biosand filters (BSFs) amended with metallic iron ($Fe^0$) to improve the contaminant removal efficiency in decentralized drinking water treatment systems.

Knowledge on designing "$Fe^0$-amended BSFs" ($Fe^0$-BSFs) has been continuously made available in the peer-reviewed literature since 2000 [11–14] and has been reviewed and updated by Noubactep and colleagues since 2009 [15–18]. Over the past decade, a science-based conceptual approach has been presented and progressively updated by our research group to address the misconceptions and controversies pertaining to $Fe^0$-filters [16–27]. For brevity, interested readers are referred to the cited literature for detailed discussions of the subject. The present communication aims to update the state-of-the-art knowledge based on the recent results of filter designs [18,28] and results characterizing the intrinsic reactivity of $Fe^0$ materials [29–33]. In this paper, first the physico-chemical properties of $Fe^0$ filters will be summarized. Then, critical considerations for designing the next generation of sustainable $Fe^0$-BSFs based on recent advances in $Fe^0$ technology will be discussed.

## 2. Conventional Biological Sand Filter (BSF)

Slow sand filtration (SSF) is the oldest engineered process for a large-scale drinking water supply [34]. The first plant based on the technology was developed by James Simpson in 1829 for the London water supply [7,35,36]. This technology was used to defeat cholera and other water-borne diseases long before the nature of the pathogens was established [7]. The most prominent example is the cholera outbreak in Hamburg and Altona in 1892 and 1893, respectively, as reported by Dr. Robert Koch [37]. There were more than 7500 deaths in Hamburg, while the neighboring Altona was nearly unaffected. Both cities got their drinking water from the same river (Elbe), but Altona is located downstream of Hamburg. Altona's water was treated using SSF, while Hamburg's water was only treated by sedimentation [7,35,37]. In recent years, the SSF has been successfully down-scaled to meet the needs of individual households and small communities [6,38,39]. The resulting device is termed a biological sand filter (BSF).

A BSF is a gravity-driven, affordable, and applicable technology for decentralized water supply. It is very effective in removing particles and pathogens by a combination of biological, chemical, and physical processes [6,7,35,36,38–44]. The operation of BSF requires no energy, and the filter's efficiency relies on the development of a biofilm (Schmutzdecke) approximately 3.0 cm below the resting water column (Figure 1). The Schmutzdecke is a biologically active mat that develops within about six weeks (maturation time). In the Schmutzdecke, nutrients and organic matter are digested and pathogens are removed by predation. Bacteria escaping the Schmutzdecke into underlying layers die because of the lack of nutrients and oxygen, while viruses escaping die because of the lack of bacteria. It is very important to underline that a BSF is an $O_2$ scavenger, while the Schmutzdecke needs $O_2$ to be formed.

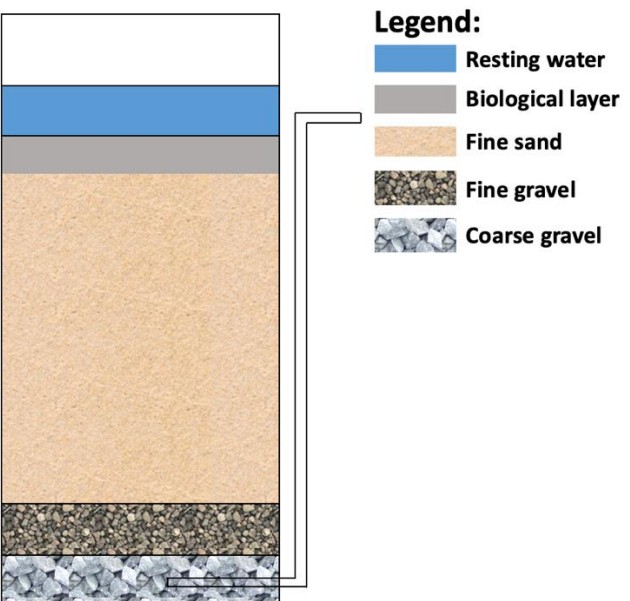

**Figure 1.** Schematic diagram of a conventional biological sand filter. The resting water column is 5.0 cm thick and the biofilm is formed within the upper 3 cm of the fine sand layer. Ideally, the sand layer is at least 50 cm thick.

All BSFs require regular operation and maintenance, including scraping off the top 2 to 5 cm of sand, changing the sand, and controlling the water flow velocity [6,39]. This simple maintenance is unfortunately not performed in many locations, and BSFs have been reported to perform poorly [45]. This situation calls for new designs aimed at reducing the maintenance workload. In other words, a filter with limited (or no) need for maintenance for a reasonable timeframe (e.g., 6 months) would be a good alternative to the conventional BSF. Reducing the maintenance and labor requirements is particularly critical in most low-income countries, including those in Africa. In such countries, the burden of household drinking water provision is often the responsibility of women and children.

In summary, the BSF is a pathogen and $O_2$ scavenger, whose functionality and performance in low-income communities are afflicted by the lack of a culture of maintenance. However, even well operating BSFs still release water containing pathogen levels that can be harmful to humans [6,39,45]. Moreover, BSFs have a limited capacity to remove dissolved contaminants such as arsenic, fluoride, and uranium, which are toxic to humans [46]. Thus, there is an urgent need to optimize the functionality of the conventional BSF and improve its capacity to remove dissolved contaminants (chemicals). One alternative is amending the BSFs with metallic iron ($Fe^0$), which has been demonstrated to effectively remove various types of contaminants for a prolonged period [12,39,47–51].

## 3. Conceptual Designs of $Fe^0$-BSF Multi-Barrier and Multi-Stage Systems

### 3.1. Fundamental Physico-Chemical Reactions in the $Fe^0/H_2O$ System

During the past 25 years, the effectiveness of metallic iron ($Fe^0$) in removing a large variety of contaminants from aqueous solutions has been demonstrated [18,27,48,52–54]. $Fe^0$ is a low-cost and readily available material worldwide, for example, in granular form or as steel wool. At pH > 4.5, $Fe^0$ corrosion generates solid iron hydroxides and oxides (iron corrosion products or FeCPs), which are good contaminant scavengers [55,56]. In essence, iron hydroxides and oxides are secondary and ternary corrosion products, while primary corrosion products are $Fe^{II}$ species and $H_2$. Since 2000, there have been some successful applications of $Fe^0$ filtration systems for drinking water systems for households [57–59] and small communities [60,61].

In the literature, $Fe^0$ is generally perceived as a reducing agent mediating the transformation of reducible contaminants according to an electrochemical mechanism [54,62]. This view has been proven false in subsequent studies, and an alternative concept has been presented in a number of papers since 2008 [63–66]. The alternative concept states that $Fe^0$ is corroded by water in an electrochemical reaction (Equation (1)), while contaminants as well as dissolved $O_2$ are reduced by primary and secondary iron corrosion products (Equation (2)) [67].

$$Fe^0 + 2\,H^+ \Rightarrow Fe^{2+} + H_2 \tag{1}$$

$$4\,Fe^{2+} + O_2 + 4\,H^+ \Rightarrow 4\,Fe^{3+} + 2\,H_2O \tag{2}$$

For the design of $Fe^0$ filters, it suffices to consider that, at pH > 4.5, each $Fe^{2+}$ from Equation (1) is transformed to iron hydroxides and oxides that are each at least twice as large in volume compared to the parent metallic iron [68,69]. The increase in the volume of iron hydroxides and oxides (contaminant scavengers) due to rusting relative to the parent iron is measured by the volumetric expansive coefficient [69]. The volumetric expansive coefficient of $Fe^0$ corrosion ranges from 2.1 to 6.4, with large values corresponding to oxic conditions, while low values relate to low or no $O_2$ [69]. In this regard, iron corrosion presents a dilemma in the design and operation of $Fe^0$ filters; on the one hand, FeCPs are beneficial for contaminant removal as scavengers, but on the other hand, they are undesirable pore-filling species which cause the loss of porosity and permeability in a $Fe^0$ filtration system [20,21,70]. The simplest tool to delay the porosity loss is to mix $Fe^0$ with non-expansive inert or reactive aggregates like gravel, manganese oxide (reactive), pumice (reactive), or sand. Thus, the design of a sustainable $Fe^0$/sand filter needs an appropriate balance between increased contaminant removal efficiency (more $Fe^0$, less sand) and sufficient porosity and permeability (more sand, less $Fe^0$). Whatever the optimal $Fe^0$/sand ratio, it is certain that operating the system under anoxic conditions (low or no $O_2$) would result in a more sustainable system than under oxic conditions. Because a BSF is an $O_2$ scavenger, a $Fe^0$ filter functions better when it follows a BSF. This is a key consideration in designing multi-barrier systems including both $Fe^0$ and BSF.

## 3.2. Amending a BSF with Metallic Iron ($Fe^0$-BSFs)

Considerable research has been undertaken in the last two decades in designing efficient $Fe^0$-based filters for low-income communities. The largest body of research was conducted as part of efforts to solve the arsenic crisis in South-East Asia and Latin America [14,58,59,61,71–73]. From a design perspective, the current filters presented in the literature can be classed into two sub-groups based on where the As removal occurs: (i) As is removed before the BSF (e.g., Kanchan filter) [12], and (ii) As is removed after the BSF (e.g., SONO filter) [57]. Both filter types were independently developed in the early 2000s. Subsequent independent research has established the superiority of the SONO design relative to the Kanchan filter and conventional BSFs [16,17,28,73,74]. Noubactep et al. [17] have discussed the processes accounting for the better efficiency of the SONO design relative to the Kanchan filter and the conventional BSF based on Figure 2.

Figure 2 gives a schematic diagram of a conventional BSF and its possible amendments with a $Fe^0$ layer. There are basically two design possibilities relative to the conventional BSF (Figure 2a): (i) the $Fe^0$ layer precedes the BSF (Figure 2c), or (ii) the BSF precedes the $Fe^0$ layer (Figure 2b,d). An $Fe^0$ layer preceding the BSF corresponds to the Kanchan filter and should be regarded as an inappropriate design layout for the BSF operation, given that $O_2$ is needed for the biofilm formation (about 8 cm under the water level). Thus, in Figure 2c no biofilm formation occurs because in this case, the $Fe^0$ layer acts as an $O_2$ scavenger. To be precise, the Kanchan design should not be termed as a $Fe^0$-BSF because the BSF component cannot properly operate as per design principle. It then follows that any quantitative contaminant removal in a Kanchan filter results from the action of the $Fe^0$ layer [17], while the BSF plays a limited or no role. A key feature of this design is that the preferential flow created in the entrance zone ($Fe^0$ layer) impairs the interactions of contaminants with the aggregates in the BSF

(fine sand, coarse sand and gravel). Conversely, low interactions between the contaminants and the aggregates lead to fewer contaminant removal opportunities by the various mechanisms. It is thus not surprising that the Kanchan filter has failed as a rule [28,72,74]. It should be explicitly stated that the design of Banerji and Chaudhari [60] corresponds to the Anderson Process, while the one discussed here corresponds to the Bischof Process [47,75]. The Anderson Process is a two-stage process in which contaminants are removed by "coagulation/flocculation" with $Fe^0$, and the flocs are then removed by sand filtration. A number of studies have also independently presented similar processes [76–78]. The presentation herein is focused on the Bischof Process, where filtration occurs through a single unit which can be composed of several compartments (Figure 3).

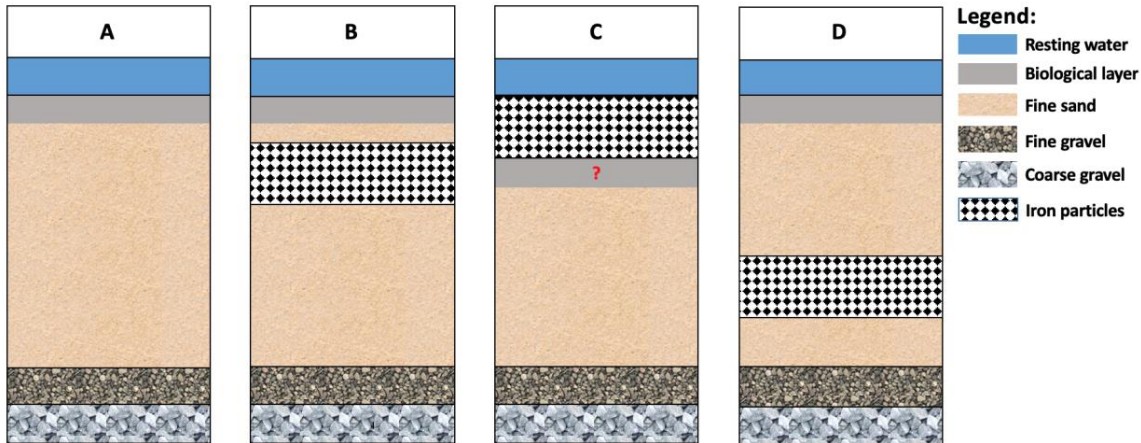

**Figure 2.** Schematic diagram of a conventional biological sand filter (**A**) and its possible amendment with metallic iron ($Fe^0$) (**B–D**). Design **C** corresponds to the Kanchan filter and suggests that the $O_2$ depletion within the $Fe^0$ layer makes the formation of the biofilm hypothetical (red line). Design **B** and **D** differ in the depth of the $Fe^0$ layer under the biofilm. The deeper the $Fe^0$ layer, the lower the $O_2$ concentration.

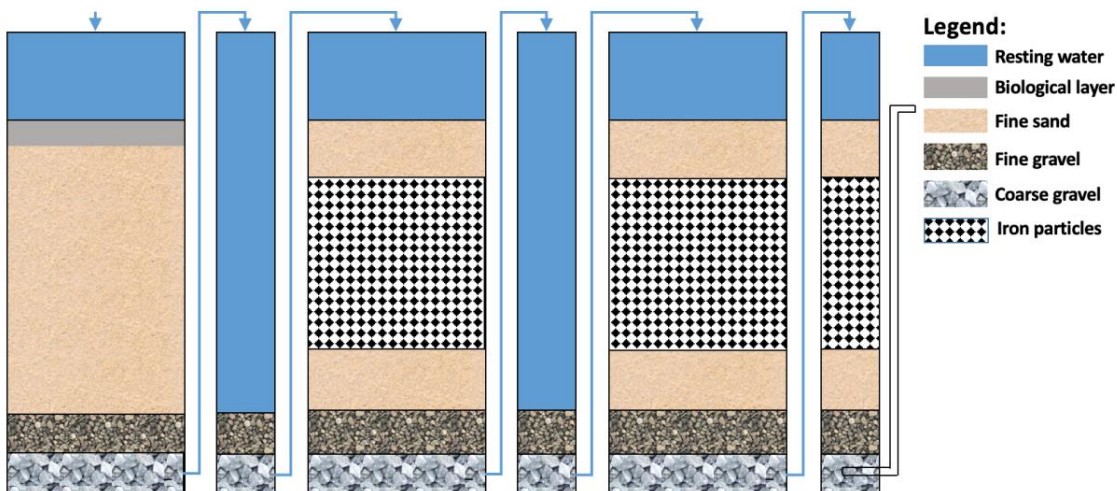

**Figure 3.** A conceptual depiction of the proposed multi-stage $Fe^0$- biological sand filters (BSF) water treatment device. From the left to the right: a biosand filter, a first equalizing filter, a first $Fe^0$/sand filter, a second equalizing filter, a second $Fe^0$/sand filter, and a filter clock.

The presentation until now demonstrates that the success of the SONO filter relies on a real synergy between the $O_2$-scavenging BSF and arsenic removal in the $Fe^0$ layer. The original SONO filter itself has been a success, and over 270,000 SONO filters were deployed in Bangladesh by 2014 [28]. On the other hand, Tuladhar and Smith [79] tested the SONO filter in rural Nepal and classified it as an "excellent technology" for safe drinking water supply. Beside arsenic and bacteria, Tuladhar and

Smith [79] documented the quantitative removal of more than 20 other contaminants in aqueous systems. The difference between designs B and D is the position of the $Fe^0$ layer within the sand layer. To maximize water decontamination by all possible mechanisms (e.g., die off, adsorption onto sand), it is better to have the $Fe^0$ layer as deep as possible (Figure 2D). This suggests that the design in Figure 2D is the optimal $Fe^0$ BSF. However, because the preferential flow initiated in the biofilm would disturb the functionality of the $Fe^0$ layer as well, the next section introduces the concept of a multi-stage system to optimize the synergy between the BSF and $Fe^0$ units.

### 3.3. The Next-Generation $Fe^0$-BSFs Multi-Stage Systems

This section develops a sound concept for efficient and sustainable compact $Fe^0$-BSFs. Much of the impetus for this conceptual design has come from the work of Tepong-Tsindé et al. [18]. The same authors successfully tested a three-unit design for pathogen and nitrate removal for one year. The tested system was made up of $Fe^0$/sand sandwiched between two BSFs while using a commercial steel wool as the $Fe^0$ material. It is expected that independent research groups will use similar designs and locally available $Fe^0$ materials to determine the system performance under relevant operating conditions (e.g., ambient temperature, nature and extent of contamination, pH value, presence of co-solutes, presence of organic matter, and turbidity).

The following intrinsic characteristics of $Fe^0/H_2O$ systems are considered: (i) all $Fe^0$ grains undergo uniform corrosion within the filter (Equation (1)); (ii) there is increased corrosion of $Fe^0$ grains in the entrance zone where oxygen and/or reducible contaminants are introduced (Equation (2)); (iii) there is likely a cake formation in the entrance zone, creating a preferential flow; (iv) the preferential flow lowers the contaminant/oxide interactions, thereby yielding early breakthrough and low contaminant removal; and (v) the preferential flow can be eliminated by incorporating an equalizing column. An equalizing column (partly) contains no solids. This is to enable the water mixing and laminar flow into the next column, thereby creating a new flow regime. The potential ability of an equalizing column to eliminate or alleviate preferential flow was observed and demonstrated in an earlier work [80].

The proposed multi-stage filtration system consists of a biosand filter (BSF) and at least one $Fe^0$-based filter in series (Figure 3). The BSF can be preceded by a screening filter for the removal of the physical contaminants (grit, turbidity). The effluent from the BSF is considerably free from oxygen ($O_2$) and pathogens. However, the residual pathogen level and the chemical contamination (if applicable) should be removed to achieve drinking water quality standards. As an $O_2$ scavenger, the BSF certainly allows the $Fe^0$ filter to operate for a longer time, as less expansive iron corrosion products are generated [20,81]. The $Fe^0$ filter inactivates pathogens and chemical contaminants promptly after implementation. This means that coupling a BSF and $Fe^0$ filter can be regarded as an efficient tool to eliminate the need to wait until maturation is achieved in the BSF in about six weeks. This evidence has already been experimentally verified by Tepong-Tsindé et al. [18], who documented quantitative nitrate and pathogens from such a system from the day of implementation to 365 days (one year). However, further research is required to test the design for a wide range of contaminants and operating conditions. As discussed by Nanseu-Njiki et al. [26], wherever necessary additional columns for the removal of specific contaminants will be added.

### 3.3.1. Controlling Preferential Flow

Iron precipitation within a $Fe^0$ filter operating at pH > 4.5 is an inherent property of aqueous iron corrosion [82,83]. Iron precipitation is beneficial for contaminant removal but contributes to the decline in the effective porosity and, in the long term, the formation of preferential flow paths [53,84–86]. A preferential flow path implies that, while the apparent flow velocity (residence time) may remain constant, there is a lower total effective adsorbing surface area for interactions with dissolved species [86]. Therefore, the monitored changes in the hydraulic conductivity of a $Fe^0$ filter alone cannot be directly related to its efficiency for water decontamination.

To better understand the impact of preferential flow paths, the time-dependent changes in a $Fe^0$ filter should be considered. A $Fe^0$ filter is initially ($t_0 = 0$) very porous and the pores are all inter-connected, like in a sand filter. As iron precipitates are in situ-generated over time ($t > t_0$), the available pores (locally) increasingly become less and less inter-connected. In the short term, there is a local clogging mainly in the entrance zone, meaning that the system is still largely porous but less permeable [18,87]. When the permeability within a $Fe^0$ filter is significantly reduced over time, preferential flow paths are formed throughout the filter. In the absence of an equalizing water column, these preferential flow paths significantly lower the effectiveness of the $Fe^0$ filter within the same water column and even in subsequent columns. Therefore, to eliminate the propagation of preferential flow from one unit (e.g., BSF or $Fe^0$/sand filter) to the next (e.g., $Fe^0$/sand filter), equalizing filters should be used (Figure 3).

In Figure 3, two equalizing filters are introduced: the first between the BSF and the first $Fe^0$/sand filter, and the second between the two $Fe^0$/sand filters. In essence, preferential flow paths are inherent to $Fe^0$ filters, such that the second equalizing filter is more important. The design of Tepong-Tsindé et al. [18] has no equalizing filter, but in some situations a single $Fe^0$/sand unit will not be capable of treating water to the required quality. Indeed, the inability of a single $Fe^0$ filter unit to treat water to meet drinking water quality standards was recently reported in $Fe^0$-BSFs tested for As removal in Burkina Faso [28]. In this case, an equalizing filter will be necessary. Following the conventional design principles for designing efficient filters [88], given polluted water and the extent of treatment needed, a compact system can be designed to achieve the water treatment goal. The number of $Fe^0$ columns in series is not limited, but depends on the initial contamination and the target treatment efficiency. The very last questions to answer are: (i) How long will a newly designed $Fe^0$-BSF last? (ii) In the case of households or communities who have limited or no access to analytical water laboratories, how can they avoid the use of exhausted filters beyond their design life? The first question can only be addressed through further research, while a conceptual design has been developed to answer the second question [89].

### 3.3.2. Avoiding the Use of Exhausted Drinking Water Filters

There are serious concerns about water quality monitoring in low-income communities worldwide, encompassing the quality of the distributed water [22,90]. In sub-Saharan Africa in particular, there are still very limited opportunities to monitor water quality because of the critical shortage of analytical equipment [90–92]. In cases where some few analytical laboratories are available, related costs represent a barrier to routine water quality monitoring for most low-income households. To protect public health, intelligent solutions are needed to avoid or limit the consumption of water of doubtful quality. To address this problem, the concept of a filter clock was recently presented [89].

The underlying principle of the filter clock is that each well-designed water filter has a defined removal capacity and service life and must be replaced when this capacity is exhausted. It may be difficult to convince some low-skilled users to buy new filters after a predicted service life. The concept of the filter-clock based on rusting iron was developed to encourage users to change their filters on time. A filter-clock contains a $Fe^0$ material whose volumetric expansive corrosion is tailored to clog promptly at the end of the service life. Such filter-clocks can be integrated to all water filters and are regarded as essential for safeguarding public health. A filter clock is the last component of the multi-stage compact unit presented in Figure 3. However, further work is required to design and evaluate such filter clocks under both laboratory and field conditions.

## 4. Future Perspectives

### 4.1. Beyond Safe Drinking Water Provision

To this point, the discussion has been limited to the application of $Fe^0$-based filters in decentralized drinking water treatment, given its criticality in safeguarding human health. However, it is

noteworthy that the concepts discussed here are also applicable for the remediation of other aquatic systems. The extension of $Fe^0$-based filters to other application domains is important, given that several anthropogenic activities, including agriculture, urban developments, and mining, generate contaminated wastewaters, effluents, and runoff [93–98].

Specifically, the concept presented herein is applicable in all technologies potentially or already using $Fe^0$-based filtration systems for water treatment. The rationale is that they work under similar conditions: (i) neutral pH values, (ii) ambient temperature, and (iii) designed to operate in decentralized manner. The main difference lies in the water chemistry—the concentrations of contaminants and major ions (ionic strength). The hitherto tested relevant technologies include: (i) river bank filtration; (ii) the treatment of domestic wastewater using multi-soil-layering technology [92]; (iii) the treatment of wetland effluents and other alternative water sources for irrigation water [93,94]; (iv) the treatment of runoff water before infiltration to increase groundwater recharge [94]; (v) the treatment of agricultural runoff, for example, to remove phosphates [95] or selenium [96]; and (vi) the treatment of mining effluents to avoid the migration of toxic species (e.g., heavy metals) into the environment [97]. Given that many of these technologies were conceived (e.g., the multi-soil-layering technology) or can be used in a decentralized manner, it appears that $Fe^0$ filtration is a veritable "one-for-all-situations" technology for clean and healthy low-income communities, including dispersed and sparsely populated rural communities in the developing world (Table 1). In other words, the $Fe^0$ filtration technology is an excellent candidate in helping the world community to achieve the UN SDG 6 even before 2030.

**Table 1.** Overview of the types of contaminants removed by $Fe^0$ filters. The selection illustrates the potential of iron filters to remove the most abundant natural pollutants: As, pathogens (bacteria and viruses) and uranium.

| Removed Species | Year | Application | Refs. |
|---|---|---|---|
| Arsenic and pathogens | 2006 | drinking water | [12] |
| Escherichia coli | 2008 | agricultural irrigation | [99] |
| Arsenic and pathogens | 2009 | drinking water | [100] |
| MS2 bacteriophage (virus) | 2011 | drinking water | [59] |
| Escherichia coli O157:H12 | 2012 | agricultural irrigation | [101] |
| Viruses and bacteriophages | 2012 | drinking water | [102] |
| Arsenic | 2013 | drinking water | [72] |
| Arsenic and uranium | 2014 | drinking water | [74] |
| Food-borne pathogens | 2017 | agricultural irrigation | [103] |
| Norovirus surrogates, murine norovirus and tulane virus | 2018 | agricultural irrigation | [104] |
| Diverse microbial communities | 2019 | agricultural irrigation | [105] |
| Antimicrobial agents | 2019 | agricultural irrigation | [106] |
| Escherichia coli and Listeria monocytogenes | 2019 | agricultural irrigation | [107] |

Table 1 recalls that $Fe^0$-based systems are efficient for the treatment of waters contaminated by antimicrobial agents [106], arsenic [12,100], bacteriophages [102], Escherichia coli [99,101], food-borne pathogens [103], Listeria monocytogenes [107], and viruses [102,104]. Additionally, Chopyk [105] investigated the diversity and the dynamics of microbial communities in wastewaters and established the capability of $Fe^0$/sand filters to reduce the concentration of all microorganisms. The diversity of the addressed contaminants confirms that adsorption, co-precipitation, and size exclusion are the fundamental mechanisms of decontamination using $Fe^0/H_2O$ systems. In other words, iron is corroded by water to generate contaminant scavengers. Table 1 also recalls that $Fe^0$/sand filters are applied in many fields, including safe drinking water supply and irrigation water.

### 4.2. Future Research

Past and current research has focused on pathogen removal by BSF and ways of amending it with adsorptive and/or reactive materials (including $Fe^0$) to extend its application to chemical contaminants. The state-of-the-art knowledge is summarized in some recent overview articles [26,39]. The compact

multi-stage filtration design tested by Tepong-Tsindé et al. [18] and using a commercial steel wool ($Fe^0$ SW) can be considered as creating a prototype for future design efforts. Several knowledge gaps still exist and include the rationale for the selection of the used $Fe^0$ materials (Table 2) and testing different arrangements of the multi-stage system (Table 3).

**Table 2.** Suggestions for future research on suitable $Fe^0$ materials for filters. The goal is to characterize the suitability of available $Fe^0$ materials for individual site-specific applications. New materials with tailored properties should be designed and tested.

| No. | Research Topic | Rationale |
|-----|----------------|-----------|
| 1 | $Fe^0$ intrinsic reactivity | There is actually no unified tool to compare/select $Fe^0$ materials |
| 2 | Characterizing steel wool | Steel wool is universally available (7 different grades). Which grades are better suitable for some applications? |
| 3 | Characterizing sponge iron | Widely available but sparely tested for $Fe^0$ filters |
| 4 | Characterizing iron wire | Iron wire should be as available as iron nails |
| 5 | Characterizing iron nails | Iron nails have been widely tested, but not really characterized |
| 6 | Characterizing scrap iron | Scrap iron can be collected from mechanical workshops |
| 7 | New $Fe^0$ materials | Designed $Fe^0$ materials could be developed and characterized |
| 8 | Compare classes of $Fe^0$ | Comparatively characterize the reactivity of $Fe^0$ |

**Table 3.** Suggestions for future research on designing compact $Fe^0$-based filters.

| No. | Research Topic | Description |
|-----|----------------|-------------|
| 1 | Equalizing units | A small filter designed to attenuate the impact of preferential flow |
| 2 | Filter clock | A small $Fe^0$ filter designed to stop flow at system exhaustion |
| 3 | Exchangeable BSF | Change the first BSF after certain intervals to avoid permeability loss |
| 4 | $Fe^0$ unit | Determine the optimal $Fe^0$/sand ratio for sustainable systems |
| 5 | Number of $Fe^0$ units | Depending on the water quality a modular approach can be adopted |
| 6 | Roughing filters (RF) | Depending on the water turbidity, RF could precede BSF |
| 7 | Multi-barrier | For contaminants insensitive to $Fe^0$ (e.g., fluoride), add new units |
| 8 | System testing | Laboratory and field experiments lasting for at least one year |

Future research should systematically test the modifications suggested herein to design more efficient and sustainable $Fe^0$ filters. Tables 2 and 3 list some relevant research topics that should be investigated. The importance of $Fe^0$ intrinsic reactivity cannot be overemphasized (Table 2). For example, each class of $Fe^0$ material listed in Table 2 should be initially characterized to determine whether there are relationships between the brand name and the efficiency for water treatment. For example, from the existing seven grades of $Fe^0$ SW, only two have been tested for water treatment in long-term experiments [18,59]. In fact, Bradley et al. [59] found that extra-fine $Fe^0$ SW (grade 0000; $d_1$ = 25 μm) was completely exhausted in a water filter after 8 months, while Tepong-Tsindé et al. [18] observed no exhaustion of fine $Fe^0$ SW (grade 0; $d_2$ = 50 μm) after 12 months. Comparing the intrinsic reactivity of several grades of $Fe^0$ SW from the same manufacturer and/or $Fe^0$ SW specimens from different suppliers would help to develop a database of possible $Fe^0$ materials for water filters for households and small communities. The same systematic work should be performed with iron filings, iron nails, sponge iron, and all other relevant materials that are readily available where $Fe^0$-based systems are needed (Table 2).

The available tests for characterizing the intrinsic reactivity of $Fe^0$ materials are limited to short-term batch experiments, which are certainly suitable for initial material screening [33,108–113]. Reliable tests to characterize the long-term reactivity of $Fe^0$ are yet to be developed [30,33,114]. If rationally selected $Fe^0$ is tested during one year and corresponding treatability tests performed for one other year (Table 3), researchers will come up in two years with an array of applicable $Fe^0$ filters.

Some of such filters will be directly transferable to the field, after being mounted in suitable filter containers to treat enough water for a typical household. The last step will be to up-scale the obtained filter devices to meet the needs of the respective households and communities. It is not superfluous to recall that $Fe^0$ filters could secure the water supply of the city of Antwerpen (Belgium) for 18 months in the early 1880s without any clogging problems [47]. The city then had 200,000 inhabitants. There is, thus, no doubt about the potential of $Fe^0$ filters to help in achieving Goal 6 of the UN SDGs even in the low-income countries.

*4.3. The Novelty of the Envisioned Fe-Based Biosand Filters*

The current paper highlighted the limitations of conventional biosand filters and highlighted a conceptual framework to guide further research. It is envisaged that addressing the current limitations and knowledge gaps highlighted will culminate in the next generation of $Fe^0$-based biosand filters. The envisaged next-generation $Fe^0$-based biosand filters will have several novelties which are currently missing in the existing designs of conventional water filters. These novelties include:

(1) A filter clock device based on the expansive nature of iron corrosion to warn the end-users when the filter reaches the end of its design service life.
(2) Flow equalizing units to enhance contaminant removal by preventing short-circuiting or preferential flow.
(3) A robust and flexible modular design that allows the incorporation of exchangeable filter units and process trains for treating raw water with a variable initial quality.
(4) An initial roughing unit for the removal of grit, turbidity, and other suspended solids that are likely to interfere with subsequent treatment processes.
(5) Multi-barrier systems, with each barrier layer tailor-made to remove a particular contaminant, including dissolved toxic geogenic pollutants such as arsenic, uranium, and fluoride.

To the authors' knowledge, no low-cost filter currently exists with the highlighted functional capabilities. In this regard, the current paper provides a comprehensive conceptual framework critical for the development of effective low-cost water filters to meet SDG 6, especially in developing countries. This is timely, given that the global community has less than 10 years to translate current scientific knowledge to frugal water treatment technologies to achieve SDG 6 by 2030. To achieve this, it is critical for the global research community working on frugal water treatment technologies to urgently conduct research anchored in the proposed conceptual framework. Such research will generate critical evidence to validate the various conceptual ideas highlighted in the current paper.

Decentralized $Fe^0$-based systems will be very useful is for the treatment of effluents from small-scale mines that are currently polluting rivers in the developing world [115–117]. In fact, well-designed $Fe^0$ systems have been used to remove heavy metals from wastewater for decades [118–121]. One recent example is the Ferrodecont process using a fluidized bed reactor [120,121].

**5. Conclusions**

$Fe^0$-amended BSFs have shown promise as an affordable, efficient, and applicable decentralized safe drinking water technology for low-income communities. Such filters are currently tested for multiple applications around the world. The innovation suggested herein is based on the physico-chemistry of the system and will certainly extend the application of $Fe^0$ BSFs, including their sustainability and their reliability. In particular, the identification of the appropriate location of the $Fe^0$ layer in the $Fe^0$-BSF design is critical in the design of the next generation of $Fe^0$ filters. Moreover, the introduction of equalizing units enables the introduction of "security" features, which are added to a functioning system for the case of reducing the risk of failure during the operation (Table 3). Such a security feature would alleviate the limitations associated with the lack of water quality monitoring, especially in low-income countries. The filter clock concept avoids the use of filters beyond their service life, an aspect that has received limited research attention. Furthermore, for contaminants such as fluoride

that are seemingly not sensitive to $Fe^0$, a compact treatment chain similar to the one presented herein should be designed. Because $Fe^0$ BSFs have been investigated for some two decades, in-depth research into long-term $Fe^0$ reactivity is the most crucial open issue. Immediate systematic investigations on this complex research subject are needed.

**Author Contributions:** Conceptualization, H.Y., C.N., R.H., A.I.N.-T., W.G., H.R.; methodology, H.Y., A.I.N.-T.; supervision, C.N. and H.R.; visualization, H.Y.; writing—original draft, C.N. and H.Y.; writing—review and editing, H.Y., C.N., R.H., A.I.N.-T., W.G., H.R. All authors have read and agreed to the published version of the manuscript.

**Funding:** This research received no external funding.

**Acknowledgments:** We acknowledge support by the German Research Foundation and the Open Access Publication Funds of the Göttingen University.

**Conflicts of Interest:** The authors declare no conflict of interest.

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
