# Peer review of "Designing the Next Generation of Fe0-Based Filters for Decentralized Safe Drinking Water Treatment: A Conceptual Framework"

_processes, doi:10.3390/pr8060745_

Round 1

Reviewer 1 Report

The authors present a review on biological sand filters combined with zero-valent iron and discuss the problem of clogging. In this context alternative concepts to address clogging such as fluidized bed reactors (ferrodecont process) should be considered as well. Apart from this the article is excellent and should be published!

Author Response

Many thanks for your evaluation. Permeability loss is addressed by admixing ZVI and sand.

We have search for "ferrodecont process" and found to articles, both considered, however under the aspect of mine water treatment.

Thanks again!

Reviewer 2 Report

The paper reviews the current knowledge of using zero-valent biological sand filters for water treatment in households and small communities and suggests future research directions for better development of scalable design. My main comment is that the use of Fe0-BSFs is highly dependent on the raw water quality and as such it would have been useful for the readers to know in a tabular form the types of contaminants that could be removed as reported by pervious studies, those that could potentially be removed but have not been studied yet, and those that cannot be removed by these systems.  Other minor comments are as follows:

Page 4, Figure 1: Present the biological layer the same as it is presented in Figure 2.

Page 5, Line 154:  Change “while contaminants, including dissolved O2” to “while some contaminants as well as dissolved O2”.

Page 8, Line 256: Change “removes” to “inactivates”.

Page 10, Line 330: Change “application in the remediation of other aquatic systems” to ”applicable for remediation of other aquatic systems”.

Page 10, Line 330: use superscript in the Fe0.

Page 12, Table 2: The equalizing units do not cancel the impact of preferential flow but maybe they reduce its impact.

Page 13, Line 411: Change “missing existing” to “missing in the existing”

Page 13, Line 413: Change “end-use” to “end-users”

Page 13, Line 426, Why “especially in developed countries” or do you mean developing countries?

Author Response

Many thanks for your time and the evaluation. We have added a table (Table 1) to account for the removable species. Il essence all species are quantitatively removed upon proper design, even fluoride but a pH shift is necessary, which is no more 'applicable' to scatter unskilled people. We have not considered this in the discussion because it makes the text long. The focus was "better design".

Many Thanks again!

Reviewer 3 Report

Dear Authors,

The work is very interesting, useful and well described.

The text is relatively easy to understand.

Authors made big and deep review.

Some remarks and questions:

What kind pathogens can to remove your described filters?

I think, it is necessary to pay more attention to the chemical composition of  water in future.  

Author Response

Many thanks for this evaluation, We have added a table specifying some pathogens. But all pathogens can be removed since the fundamental mechanism is occlusion. Adsorption and size-exclusion are also very important.

Many Thanks!

Reviewer 4 Report

This concept paper summarizes the state-up-to date knowledge on designing Fe0-BSFs filters for drinking water treatment in households and small communities. The utilization biological sand filters (BSFs) amended with metallic iron (Fe0 -BSFs) shows the progress in quantitative remove pathogens and the chemical pollutants. Available data about Fe0-BSF filters were achieved under various operating conditions. 

The well-designed laboratory and field experiments is required to increase the efficiency of Fe0-BSFs filters. Tools to alleviate permeability loss, preferential flow and the use of exhausted filters are presented in this manuscript.

The manuscript is interesting, it is writted at a very good level, it is up-to-date, original. Now, the removal of pathogen, micropolutants and various chemical compounds from waters (ground, surface, atmospheric, drinking) with different processes of water treatment are currently being investigated worldwide, and new possibilities and technologies of water treatment are constantly being sought. In the manuscript is writted about the next generation of Fe0-based filters.

I have a several questions and comments:

Why was the most frequently used sorption materials GEH, Bayoxide E33, for removing of arsenic from water not mentioned?

For removal of As, it is important oxidation conditions in the filter, AsV  is 10 times easier to remove than AsIII, how will the proposed filter ensure this?

How can a layer of ferrous material be placed between two layers of sand?

How does the operator, the consumer in a small community, get a slow sand filter - with a biological membrane created, or will it have to be created by the customer himself?

Would it not be more appropriate to place the filter with the sand with biological membrane and the filter with the ferrous material behind it in series?

You are using these filters to remove nitrates, fluorides, arsenic ... There is a big difference in removing of nitrates, fluorides and arsenic, each of them requires a different technology

Does the quality of the raw water (eg pH, temperature, chemical composition, solutes and insolubles, turbidity, organic matter, etc.) affect the operation of the filter? In developing regions and at the educational level of the population, it is inconceivable that they operate such filter themselves, resp. complete water treatment (the article talks about domestic water treatment, or for small (communities),

How do you want to ensure proper operation of the filters?

Reviewer 5 Report

This manuscript presents a concept of designing the next generation of Fe0-based filterers for decentralized, safe drinking water treatment.

  1. What is the mechanism of filtering the pathogens has not been discussed in the manuscript, please elaborate?
  2. What are the practical issues when implementing the concept in this paper?

Author Response

Many thanks for your evaluation.

  1. What is the mechanism of filtering the pathogens has not been discussed in the manuscript, please elaborate?

We have clearly stated the iron corrosion products are contaminant scavengers and we have referred to the many previous papers. In reponse to this comment, we have written that adsorption, co-precipitation and size-exclusion are the removal mechanisms.

  1. What are the practical issues when implementing the concept in this paper?

This is over the scope of this concept paper. It is written to address the misconceptions of previous designs and make suggestions for the future. One of the reviewer has pointed out that equilization unit will not fully cancel the impact of differential flow. In the context of iron filters, no investigations have been presented,

Sincerely,

Dr. Noubactep

Round 2

Reviewer 5 Report

i do not have further comments.

Author Response

Thanks for your comments